# Dangerous Stops: Nonsense Mutations Can Dramatically Increase Frequency of Prion Conversion

**DOI:** 10.3390/ijms22041542

**Published:** 2021-02-03

**Authors:** Alexander A. Dergalev, Valery N. Urakov, Michael O. Agaphonov, Alexander I. Alexandrov, Vitaly V. Kushnirov

**Affiliations:** A.N. Bach Institute of Biochemistry, Federal Research Center “Fundamentals of Biotechnology” of the Russian Academy of Sciences, 119071 Moscow, Russia; alexanderdergalioff@gmail.com (A.A.D.); valery.urakov@gmail.com (V.N.U.); agaphonov@inbi.ras.ru (M.O.A.)

**Keywords:** prion, amyloid, prion appearance, prion structure, Sup35, Rnq1, proteinase

## Abstract

Amyloid formation is associated with many incurable diseases. For some of these, sporadic cases are much more common than familial ones. Some reports point to the role of somatic cell mosaicism in these cases via origination of amyloids in a limited number of cells, which can then spread through tissues. However, specific types of sporadic mutations responsible for such effects are unknown. In order to identify mutations capable of increasing the de novo appearance of amyloids, we searched for such mutants in the yeast prionogenic protein Sup35. We introduced to yeast cells an additional copy of the *SUP35* gene with mutated amyloidogenic domain and observed that some nonsense mutations increased the incidence of prions by several orders of magnitude. This effect was related to exposure at the C-terminus of an internal amyloidogenic region of Sup35. We also discovered that *SUP35* mRNA could undergo splicing, although inefficiently, causing appearance of a shortened Sup35 isoform lacking its functional domain, which was also highly prionogenic. Our data suggest that truncated forms of amyloidogenic proteins, resulting from nonsense mutations or alternative splicing in rare somatic cells, might initiate spontaneous localized formation of amyloids, which can then spread, resulting in sporadic amyloid disease.

## 1. Introduction

Amyloids are filamentous protein aggregates with regular cross-beta structure. Amyloids are a hallmark of many degenerative diseases, such as prion diseases, Alzheimer’s, Parkinson’s and Huntington’s diseases, Amyotrophic Lateral Sclerosis (ALS), type 2 diabetes and many less common disorders [1]. Some amyloid-associated diseases are mostly familial, i.e., they have a clear genetic cause—inherited mutations present in all somatic cells. For instance, most cases of Huntington’s disease and Gerstmann–Sträussler–Scheinker syndrome are associated with inherited mutations in HTT and PRNP genes, respectively [2,3]. A small proportion of Alzheimer’s (AD) and Parkinson’s disease (PD) cases are also linked with inherited mutations in the genes of several amyloidogenic proteins [4,5].

However, the majority of AD, PD and prion diseases cases are sporadic, and little is known about the mechanisms of disease emergence in these cases. Collapse of proteostasis and amyloid-handling systems, and aberrant post-translational modifications of a causative protein [6] were hypothesized to initiate sporadic amyloidosis onset. An alternative explanation is the “somatic mutation theory”, which claims that postzygotic mutations in the causative genes might induce amyloidization of the corresponding proteins in specific cells [7]. An important detail, related to both theories, is that despite the fact that amyloidoses are mostly non-infectious between organisms, there is increasing evidence that many of them, including A-beta, synuclein and tau protein, can spread within an organism via a prion-like mechanism [8,9]. Thus, amyloid polymers appearing in rare somatic cells with pro-amyloid mutations might then spread and infect large cell populations, or certain tissues. However, it is not known, what kind of mutations might act in such a fashion. In this work, study of the yeast Sup35 prion appearance allowed us to uncover a likely and general type of mutation that can increase the likelihood of amyloid appearance. 

There are about ten proteins in the yeast *Saccharomyces cerevisiae*, which can form amyloids stably heritable in cellular generations [10,11,12]. These amyloids, as well as related phenotypes are called yeast prions for their fundamental similarity to human prions. Yeast prions represent a convenient and fruitful model for studying the basic rules of prion and amyloid formation and propagation. The best studied yeast prion is [*PSI*^+^], which is related to the amyloid form of the yeast translation termination factor eRF3, also known as Sup35 [12]. Sup35 includes the essential GTP-ase C domain and non-essential intrinsically disordered NM region, which can drive the formation and inheritance of multiple distinct amyloid folds, that underlie different variants of [*PSI*^+^] phenotype [13,14,15]. These variants can be divided into two major classes, “weak” and “strong”. “Weak” [*PSI*^+^] show lower efficiency of nonsense suppression and higher level of soluble Sup35 than the “strong” ones. Observation of the [*PSI*^+^] phenotype is facilitated by the *ade1–14* nonsense mutation. [*psi*^−^] *ade1–14* cells require adenine and form red colonies due to accumulation of red intermediate of adenine synthesis. [*PSI*^+^] reduces the level of functional Sup35 and impairs translation termination, causing readthrough of *ade1–14* and partial restoration of adenine synthesis. Such cells became adenine independent and form white or pink colonies. [*PSI*^+^] has a very low rate of spontaneous formation (estimated as 6 × 10^−7^, [16]) in normal cells with endogenous levels of Sup35 production. [*PSI*^+^] appearance with high frequency usually requires overproduction of Sup35, or its prion domain, as well as the presence of another prion, [*PIN*^+^], related to amyloid of the Rnq1 protein [17]. Presumably, [*PSI*^+^] initiation occurs through direct contact of Sup35 with Rnq1 prion particles [18]. Moreover, [*PSI*^+^] appearance is increased to some extent by various stresses and in aged cells [17,19,20]. 

Notably, little is known about mutations in the Sup35 protein, which cause increased [*PSI*^+^] formation in the absence of Sup35 overproduction. Possibly, the only known mutation of such kind is addition of two oligopeptide repeats to the Sup35 N domain that increased the [*PSI*^+^] appearance frequency up to 5 × 10^−3^ [21]. However, this mutation was made artificially, rather than obtained spontaneously. This prompted us to search for mutant variants of the Sup35 prion region that provoke [*PSI*^+^] formation by genomic wild-type Sup35 allele. We discovered two nonsense mutations in the N domain, which strongly stimulated [*PSI*^+^] formation, converting a wild-type Sup35 into prion form. In order to verify whether truncated Sup35NM isoforms universally exhibited this feature, we tested a collection of designed truncation variants with different length, and found that only a certain range of polypeptide length can induce the [*PSI^+^*] state efficiently. Surprisingly, this range correlates with subdomain organization of Sup35 protein and with the structural organization of amyloid fibers formed by full length protein, as understood by the location of proteinase K (PK)-resistant cores in Sup35 amyloid fibrils [15]. We also cloned the earlier predicted [22] spliced Sup35 mRNA encoding the N domain (1–122) plus a short fragment translated out of frame, which also showed high prionogenicity when expressed from a centromeric plasmid. In sum, our data suggest that truncated forms of amyloidogenic proteins originating from spontaneous nonsense mutations or splicing can exhibit highly increased amyloidogenic potential and might thus play an important role in sporadic cases of amyloid diseases.

## 2. Results

### 2.1. Isolation of SUP35 Mutants That Induce [PSI^+^] without Their Overproduction

We searched for mutations in the prion (N) domain of the Sup35 protein that would yield frequent [*PSI*^+^] appearance at standard Sup35 expression level. For this, the DNA fragment corresponding to the N domain was amplified by mutagenic PCR, and then inserted to the centromeric YCplac111-SUP35NM plasmid by homologous recombination in yeast (Figure 1). Transformants were plated to a medium with reduced adenine content, where [*psi*^−^] colonies are red due to *ade1–14* nonsense mutation, while [*PSI*^+^] cells produce pink or white colonies indicating suppression of this mutation. About 30 transformants with [*PSI*^+^] phenotype were obtained; plasmids were isolated from them and re-introduced into the same strain. In most cases, the plasmids did not cause frequent [*PSI*^+^] appearance, except for two, which gave sectoring colonies after about seven days of growth. Sequencing of these plasmids revealed that both contained nonsense mutations, located at positions 101 or 110, as well as some missense mutations (Q38R, Y45C, Y101Stop and N100H, L110Stop). This led us to suggest that the cause of the effect was production of truncated Sup35. 

### 2.2. [PSI^+^] Induction by Sup35 N-Terminal Fragments

To check this idea, we created a set of *SUP35* constructs, based on centromeric plasmids and using native *SUP35* promoter, encoding varying length of the Sup35 protein, ending at positions 18, 37, 49, 64, 74, 83, 93, 101, 112, 123 and 252. These Sup35 N-terminal fragments were produced in a [*PIN*^+^] background. Subsequent analysis of [*PSI*^+^] appearance showed that its frequency depended strongly on the Sup35 fragment length (Figure 2, Table 1). The highest frequency, 8 × 10^−2^, was observed for the 1–112 fragment (Figure 3). This value was comparable to those obtained with multicopy *SUP35* overproduction (10^−2^, [18]) and exceeded such frequency for an additional copy of full Sup35 by ~6000-fold. Slightly lower frequencies (2–4 × 10^−2^) were observed for the original mutants. For all *SUP35* constructs, the [*PSI*^+^] appearance strongly depended on [*PIN*^+^], being lower than 2 × 10^−5^ in [*pin*^−^] transformants.

Propagation of [*PSI*^+^] prions, obtained in such way, did not depend on the inducing plasmid encoding truncated Sup35: its loss did not cure [*PSI*^+^] in any of the studied cases (Appendix A).

### 2.3. Structure of the Inducer Fragment

Since truncated Sup35 was the cause of increased appearance of [*PSI*^+^], it is reasonable to suggest that the prion conversion was started by truncated Sup35 and then passed on to the full-length protein. The Sup35 N domain can form two PK-resistant prion structures, Core 1 (residues 2–72) and Core 2 (91–121) [15]. Core 1 was found in all [*PSI*^+^] variants, while Core 2 in about half of them, mainly in “weak” [*PSI*^+^] variants with lower efficiency of nonsense suppression [15]. Core 1 appears to be more important in all respects: it defines the [*PSI*^+^] phenotype; it is sufficient to aggregate Sup35 and transmit the [*PSI*^+^] variant-specific amyloid fold; and all known anti-prion mutations are located in the Core 1 [15,23,24]. 

However, Core 2 could play the key role in the effects described here. Since the only difference between the truncated Sup35 proteins was their C terminus, we assumed that the increased frequency of prion appearance was related to the rapid formation of amyloid structure of Core 2 when it is located at the C terminus. The high amyloidogenic potential of this region is also predicted by the WALTZ algorithm, according to which the stretch 98–118 is the most amyloidogenic part of Sup35 [25]. To test this idea, we studied the structure of Sup35(1–112) amyloid, which causes the highest frequency of prion conversion. To avoid the influence of full Sup35, Sup35(1–112) was overproduced in the 9A-H67 [*PIN*^+^] strain harboring a chromosomal deletion of the region coding for the Sup35 N and *M* domains. Sup35(1–112) amyloid was isolated, digested with PK and the resistant peptides were analyzed by MALDI and interpreted as described [15]. The derived structure included both Core 1 (residues 2–72) and Core 2 (~62–112) (Figure 4). The obtained profile did not prove the primary role of the Core 2, but it did not contradict such a role, since amyloid formation at Core 2 was expected to cause amyloid folding of the Core 1 region.

Importantly, Core 2 structure of Sup35(1–112) showed a significant difference with Core 2 variants observed in Sup35NM-GFP [15]. It has a relatively large region (92–112) fully protected from PK. This reflects an observation that no peptides ending within this region were found after thorough investigation of the MALDI spectrum. A fully protected region was observed very rarely in Core 2 variants of Sup35NM-GFP. Besides, Core 2 of Sup35(1–112) was shifted about ten residues towards the N-terminus compared to standard Core 2 location [15]. Thus, Core 2 of Sup35(1–112) appears to be a different structure. It bears some similarity to Core 1: their terminal sides are protected fully, while the inner parts are protected only partially. 

The Core 1 spectrum of Sup35(1–112) amyloid was typical of a weak [*PSI*^+^] variant, as indicated by the high relative abundance of peptide 2–42 compared to 2–35 and 2–38 [15]. In agreement with this, almost all appearing [*PSI*^+^] colonies showed the same weak [*PSI*^+^] phenotype (Figure 2, bottom row).

We also mapped proteinase K-resistant prion structures for two [*PSI*^+^] isolates with a dark pink phenotype most abundant among prions induced by the 1–101 and 1–110 Sup35 mutants (Appendix A). The latter was a typical weak [*PSI*^+^] structure. The former was also weak, but was somewhat unusual, since its protected structure ended at residue 42 and Core 2 was absent.

### 2.4. SUP35 Splicing 

Our results conclusively show that truncation of Sup35 (such as that obtained via nonsense mutations in the prion domain) can strongly increase the frequency of prion emergence. What other scenarios could provide truncated Sup35, and thus increase the rate of prion formation? We suggested that one such scenario could be splicing, which was proposed by our group in 1988 based on *SUP35* sequence analysis (Figure 5) [22]. *SUP35* includes perfect consensus sequences for a 5′ splice site (GTATGT) and lariat sequence (TACTAAC), though the distance between them, 1330 bases, is significantly larger than the average size of yeast introns [26]. However, the possibility of splicing at these sites was not studied. Now, we were able to clone the predicted splicing product from yeast cDNA. We had to use a shortened PCR cycle to restrict amplification of the full-sized *SUP35*, which indicates that the proportion of spliced mRNA was low. Sequencing of the cloned DNA confirmed that intron excision occurred exactly by the rules of splicing. A minor difference with the earlier prediction was that the 3′ splice site was not the first TAG after the lariat, but the first AG, which better fits the modern understanding of splicing [27]. The splicing product, designated as Sup35N-Spl, includes the Sup35 N domain (1–122) fused to 36 residues translated out of frame from the *SUP35* C region. Thus, Sup35N-Spl cannot perform the essential function of Sup35 in translation. Though the proportion of spliced *SUP35* mRNA appears to be very low, it cannot be ruled out that under some yet unknown conditions this proportion can be significant. 

The *SUP35N*-*Spl* gene on a low copy pRS316 plasmid caused increased [*PSI*^+^] production at a frequency of 3.7 ± 1.5%, which is comparable to the most efficient truncated Sup35 variants. [*PSI*^+^] appearance in this case also depended on the [*PIN*^+^] prion. 

## 3. Discussion

### 3.1. The “Hot-Spot” of Sup35 Prionogenesis and Its Possible Structural Explanations

In the present work, we observed that some nonsense mutations within Sup35 prion domain can increase the frequency of [*PSI*^+^] appearance by four orders of magnitude, providing [*PSI*^+^] appearance rates close to those of Sup35 or Sup35NM overproduction. Notably, the highest increase in [*PSI*^+^] appearance was observed for Sup35 fragments ending in a relatively narrow region (84–112) of the prion domain. This region corresponds to oligopeptide repeats and the second proteinase K-resistant prion structure in Sup35 (Core 2).

Why does C-terminal shortening of Sup35 increase the prion appearance rate so dramatically? The presence of the large functional C domain can be one of causes. [*PSI*^+^] appearance depended on [*PIN*^+^], so apparently, and in line with earlier observations [18,28], interaction of Sup35 molecule with aggregated Rnq1 was a key step in this process. The C domain can reduce the probability of such interaction by shifting Sup35 localization towards ribosomes, while aggregated Rnq1 is predominantly located at the perivacuolar insoluble protein deposit (IPOD) [29]. However, the difference in [*PSI*^+^] appearance, attributable to the C domain, i.e., the one between the full length Sup35 and Sup35NM is only four-fold, while the difference between Sup35NM and Sup35(1–112) is 1600-fold, and even the difference with Sup35(1–123) is 27-fold (Table 1). Since the cellular location of Sup35 seems not to be very important, the next likely explanation is that the increased frequency of prion appearance was related to the rapid formation of amyloid structure of Core 2 when the respective amyloidogenic sequence was located exactly at the C terminus.

This explanation has an important recent precedent. The Cyc8 protein can form a prion due to inner glutamine-rich disordered region [30]. Dysfunction of the Cyc8 protein allows yeast to utilize mannitol and sorbitol as sole carbon sources. Tanaka and co-authors [31] observed that for some nonsense mutations in the *CYC8* gene the ability to utilize mannitol and sorbitol is sensitive to guanidine hydrochloride and requires *HSP104*. These are characteristic tests for prion formation. They strongly suggest that truncated Cyc8 readily converts to heritable amyloid (prion), though this was not directly shown in this work. Similar to our case, all such mutations were located near the end of a prionogenic region, in this case, a polyglutamine tract. 

Thus, our case and that of Cyc8 suggest that the propensity of amyloidogenic regions to acquire amyloid fold de novo depends on their location within a protein and becomes maximal, when a region is located at the terminus. Then, the presence of non-prionogenic sequences after such regions should strongly reduce the frequency of prion conversion. 

The case of Cyc8 likely reveals another phenomenon: the amyloid fold of C-terminally truncated Cyc8 appears to be incompatible with the full-sized protein, since an additional copy of full-sized *CYC8* reverted the prion phenotype [31]. A similar effect is observed for the Sup35 N-terminus: joining of the glutathione transferase to it blocks its abilities to either interact with preexisting Sup35 prion [32] or to form a prion on its own (our unpublished data). While the prion fold was readily transferred from C-terminally truncated Sup35 to full Sup35, we consider it likely that the transfer occurred through the N-terminal Core 1, rather than Core 2. Finally, many human amyloids, for example, Aβ, are composed of protein fragments, while full proteins are not included. 

The described examples suggest that the prion fold acquired by the terminally located prionogenic region may not be accepted by the same region located inside of a polypeptide. In support of this, our data indicate that the Sup35 Core 2 region acquires different amyloid folds, when located at a terminus or inside of the full protein. At internal location, Sup35 Core 2 is only partially protease resistant in most of the studied [*PSI*^+^] variants [15]. In contrast, in the Core 2 of Sup35(1–112) the C-terminal region (92–112) was fully protected from PK. Interestingly, full protection of the terminus of an amyloidogenic region is shared by all terminal prion cores whose structure was studied so far with PK. These include: the Sup35 N-terminal Core 1 in 26 [*PSI*^+^] isolates and the C-terminal core of Rnq1 in 15 of the same isolates, where Rnq1 was present or clearly visible [15], as well as the N-terminal cores of Mot3 and Asm4 amyloids (manuscript in preparation). 

We proposed earlier, based on genetic data and PK digestion [23], that the N-terminus of Sup35 is likely to be buried inside the amyloid structure (Figure 3). In such a structure it may be difficult to accommodate any additional polypeptide chain added to the terminus, and so this fold would not be transferable to the same sequence inside of a polypeptide. Such logic can apply equally to N- and C-terminally located prion cores. In contrast, in a popular theoretical model for Sup35 prion structure, which proposes serpent-like chain folding [33], no such inhibitory effects should be expected (Figure 6). 

Altogether, these observations suggest that amyloidogenic regions acquire amyloid fold de novo much faster, when located near the polypeptide terminus (N- or C-), though these folds may transfer poorly to the same region inside of a larger protein. This conclusion could explain why many human amyloids are composed of protein fragments, rather than full proteins. 

### 3.2. Sup35 Splicing and Its Effect on Prionogenesis

The mechanisms considered in the previous chapter cannot explain the increased [*PSI*^+^] appearance by Sup35N-Spl resulting from splicing. Sup35N-Spl is 35 residues longer than Sup35N, but shows much higher prion appearance. In this case, another mechanism is more likely. Earlier it was shown [19,34] that short peptides attached to various Sup35 N-terminal fragments due to lack of proper stop codon and translation of non-coding sequences, called “magic tails”, significantly increase frequency of [*PSI*^+^] appearance and decrease dependence on [*PIN*^+^]. Presumably, such peptides are unstructured and tend to aggregate, which promotes prion formation by the Sup35 N-terminal region. In Sup35N-Spl, the sequence downstream of the splice site also represents a random “magic tail”, since it was translated out of frame. 

Notably, a more common way of obtaining protein truncations, combined with “magic tails”, is not splicing, which usually preserves the frame, but frameshift mutations. As an example, deletion of two base pairs in codon 178 of human PRNP gene resulted in a tail of 25 amino acids translated out of frame [35]. This mutation was associated with sensory neuropathy and cognitive impairment. Another 2-bp deletion, at codon 126 of Cu/Zn superoxide dismutase SOD1 is associated with familial ALS [36].

Alternative splicing in man usually is not supposed to alter the frame and therefore should not generate “magic tails”. Nevertheless, splicing can cause amyloidosis: shortened TDP43 isoforms resulting from alternative splicing trigger massive TDP43 amyloidization involving full-length protein and drive ALS pathology [37].

### 3.3. Support for the “Somatic Mutation” Hypothesis of Sporadic Amyloidoses

The described findings may also help understanding emergence of sporadic amyloidoses. According to the “somatic mutation” hypothesis of genesis of sporadic amyloid diseases [7], amyloids can appear in single cells carrying certain postzygotic mutations and then spread to new cells and tissues via prion-like mechanism [9]. Our study shows that one kind of such mutations are nonsense mutations, and they can cause a dramatic increase in the frequency of amyloid appearance. Consistent with our data, a number of inherited nonsense mutations in PrP [35,38,39], FUS [40,41]; and SOD1 [42,43], were earlier associated with various neurodegenerative pathologies.

How many cells in a human body could carry somatic nonsense mutations at hot-spots for amyloid formation? The somatic mutations frequency in humans is about 3 × 10^−7^ per base pair [44], and this value can also be regarded as a probability of mutation at any given position in the genome. The human body contains almost 10^14^ cells, and so there must be about 10^14^ × 3 × 10^−7^ = 3 × 10^7^ cells with a mutation at any given nucleotide. This simple calculation would be correct for a haploid genome, but since the genome is diploid, the probability of mutation in just one of two gene copies is twice as high. Then one should consider that not every mutation is a nonsense mutation, but on the other hand, the hotspot regions can span for tens of base pairs, which together slightly increases the estimate. Thus, one should expect about 10^8^ cells with nonsense mutations at hot-spots for amyloid formation for any type of amyloidosis. So, the presence of such cells does not seem to be a limiting factor for onset of amyloidosis. 

What other restrictions could exist for amyloid development, and thus be potential targets for its prevention? One is that the amyloid fold, acquired by a truncated protein, could be incompatible with a full protein, as discussed in Section 3.1. This effect can vary significantly depending on the particular protein. There is currently no understanding of the rules governing such compatibility, and the yeast prion model seems promising for studying this issue. Another restriction comes from the observation that prion formation by truncated Sup35 strongly requires presence of another prion, [*PIN*^+^]. In general, amyloids promote appearance of each other, though strength of the effect may depend significantly on participating proteins [28,45,46] The initiating “helper” amyloid(s) could lack an evident manifestation, like the yeast [*PIN*^+^], and so it may be difficult to identify them. However, at least one amyloid is present in almost all persons older than 60: the medin, an internal fragment of lactadherin [47]. Increased levels of medin are associated with Alzheimer’s disease [48]. Finally, propagation of amyloid within human tissues is a complex process with multiple mechanisms (reviewed in [49], which is likely to vary greatly in its efficiency depending on both the specific protein and its amyloid fold. Study of this process may be helped by humanized cellular models of amyloid spreading, such as 3D self-organized systems built from induced pluripotent stem cells [50,51,52].

## 4. Materials and Methods

### 4.1. Yeast Strains and Media

Yeast strains 74-D694 [*psi*^−^] (*MATa ade1–14 ura3–52 leu2–3,112 his3-D200 trp1–289*) harboring [*PIN*^+^], [53] and 9A-H67 [*PIN*^+^] (*MATa ura3–52 leu2–3,112 his3 ade2–1 SUQ5 SUP35C cyhR*) were used. Synthetic complete (SC) media contained 6.7 g/L yeast nitrogen base, 20 g/L glucose or galactose, and required amino acids. For colony color development, SC media contained reduced amount of adenine (7 mg/L, or 1/3 of standard). Adenine-limiting rich medium (YPDred) contained 5 g/L yeast extract, 20 g/L peptone, 20 g/L glucose and 20 g/L agar.

### 4.2. Plasmids 

For *SUP35* mutagenesis, the plasmid YCplac111-SUP35NM (cen, *LEU2*) was used, which encodes first 252 amino acid residues of Sup35, followed by 2HA tag, under the *SUP35* promoter. In this plasmid, the codon 124 was mutagenized from ATG (Met) to GCC (Ala) to create an *Ehe*I site (GGCGCC). 

Truncated *SUP35* constructs were mainly produced from the set of plasmids with deletions in the *SUP35 N* domain [54]. These deletions have a common 3′ end at the natural *Eco*RV site at residue 112. We deleted a fragment from this site to the *Hpa*I site in the C domain-coding region, which created an in-frame stop codon. This resulted in plasmids coding for the N-terminal Sup35 fragments ending at positions from 19 to 112. DNA encoding the full N domain (1–123) was amplified by primers GCGGTTTCTTCATCGACTTG and GTTAACCTTGAGACTGTGGTTGGA. The latter is a reverse primer adding TAA stop codon after glycine-123. The PCR product was cut with *Kpn*I and the downstream of the two fragments was cloned into Yeplac111-SUP35NM cut with *Kpn*I and *Bal*I.

To obtain “spliced” *SUP35* gene, cDNA was synthesized using as a template poly A+ RNA isolated from 5V-H19 strain. The specific product was obtained by PCR with primers GCCCAACCTGCAGGTGGG and ACAGCGATGACCTTCATACC. Short elongation time (10 s) was used to block amplification of the nonspliced message. The 450 bp PCR product was sequenced. The internal EcoRV-Cfr10I fragment of the wild type *SUP35* in pRS316 was replaced with corresponding *Eco*RV-*Cfr*10I fragment of the PCR product. 

### 4.3. SUP35 Mutagenesis and Obtaining Mutants with Increased [PSI^+^] Appearance 

The *SUP35* region encoding the N domain with small flanks of the *M* domain and *SUP35* promoter was amplified from the YCplac111-Sup35NM-HA plasmid using Sup35D1 (CACTCGACCAAAGCTCCCA) and Sup35R1 (CCTTCTTGGTAGCATTGGC) primers. PCR reaction was performed for 18 cycles at standard conditions, followed by 2 cycles of error-prone PCR in a separate reaction with Taq polymerase, 10x elevated concentrations of dATP and dGTP, 7 mM MgCl_2_ and 0.5 mM MnCl_2_). 

The plasmid YCplac111-SUP35NM was cut with KpnI and EheI and used to transform 74-D694 [*psi*^−^] [*PIN*^+^] yeast together with mutagenized *SUP35N* fragment. Transformants were plated to a medium with reduced adenine content to reveal the [*PSI*^+^] phenotype. Subsequent procedures are described in the text.

### 4.4. Evaluation of PSI^+^ Induction Rates

74-D694 [*psi*^−^] [*PIN*^+^] cells were transformed with centromeric plasmids encoding truncated Sup35 variants and plated to solid SC media, selective for plasmids and with low (7 mg/L) adenine. After 4 days of growth at 30 °C, individual colonies of ~2 mm in diameter were suspended in1 mL of water. 30 μL of this suspension were plated to SC medium lacking adenine for counting Ade+ cells, and 1 μL was spread on YPD plate for counting a total number of viable cells. The resulting colonies were counted using OpenCFU software [55]. For each Sup35 mutant, from 3 to 5 individual transformant colonies were treated in this way, as independent replicates, and mean proportion of Ade+ cells ± SEM was calculated. To prove that Ade+ phenotype was due to [*PSI*^+^] prion, several tens of clones were streaked on YPD-red plate with 4 mM Guanidine hydrochloride, and the change of colony color to red was monitored.

### 4.5. PK Digestion, Mass Spectrometry an Analysis

Sup35NM-GFP and Sup35(1–112) were overproduced, isolated and analyzed as described [15]. Briefly, 4 μg of purified amyloid was digested by PK (25–100 μg/mL) in 20 μL for 1 h at room temperature, PK was inactivated by PMSF and the reaction was precipitated by 16 µL of acetone, washed with acetone, boiled for 3 min and analyzed by MALDI-TOF/TOF mass spectrometer UltrafleXtreme (Bruker, Germany). Peptides were identified by MS-MS and/or as groups of related peaks.

To graphically represent the PK resistant structures, we calculated for every residue of Sup35NM the PK resistance index as a sum of mass spectral peak areas of peptides which include this residue. The resulting graphs were normalized against their maximum value for each preparation.

## Figures and Tables

**Figure 1 ijms-22-01542-f001:**
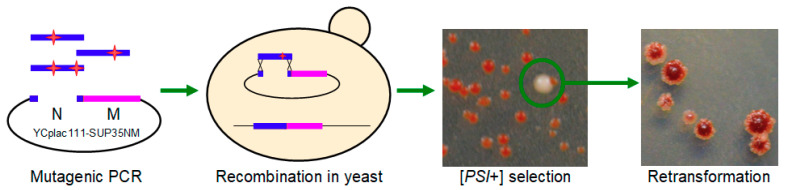
Isolation of Sup35N mutants able to cause [*PSI*^+^] appearance without overproduction.

**Figure 2 ijms-22-01542-f002:**
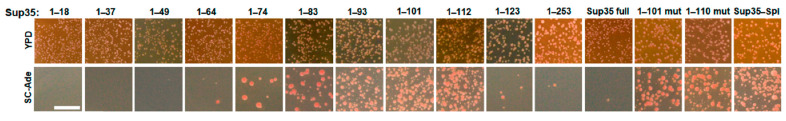
Frequency of the [*PSI*^+^] appearance caused by the tested Sup35N proteins. Yeast were plated as single cells to YPD medium or Sc-Ade, Adenine omission medium selective for [*PSI*^+^] cells. 30-fold more cells were plated to Sc-Ade medium. The scale bar is 1 cm. Sup35 variants are indicated on top.

**Figure 3 ijms-22-01542-f003:**
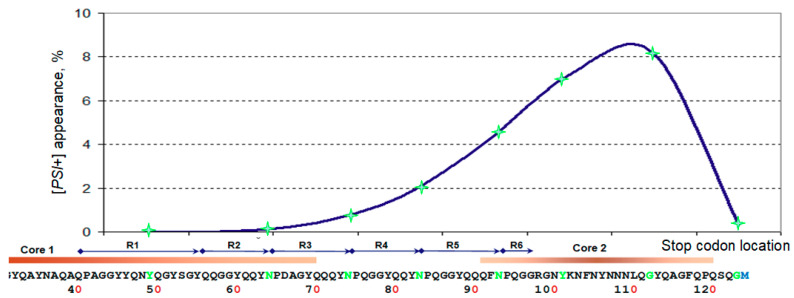
Dependence of the [*PSI*^+^] appearance frequency on the location of an inducer C-terminal residue (green) on the Sup35N sequence. The locations of oligopeptide repeats and prion Cores 1 and 2 typical for weak [*PSI*^+^] variants are shown.

**Figure 4 ijms-22-01542-f004:**
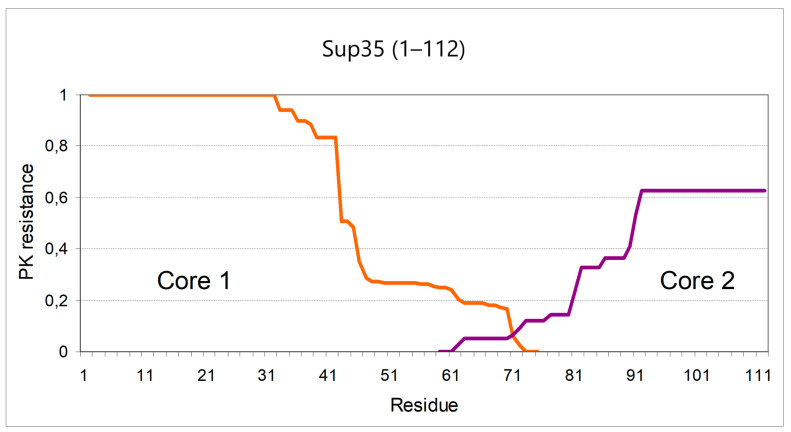
Proteinase K resistance profile of the Sup35(1–112) amyloid. The PK resistance index was calculated for every residue of Sup35NM as a sum of mass spectral peak areas of resistant peptides which include this residue [15]. The orange line sums the peptides starting from residue 2 (Core 1. First methionine is removed in yeast); purple line sums the peptides ending at residue 112 (Core 2). No other peptides were found. The terminal parts of the Core 1 and Core 2 are fully protected and therefore are equally resistant to PK. The difference in PK resistance values only reflects variation of the MALDI procedure.

**Figure 5 ijms-22-01542-f005:**
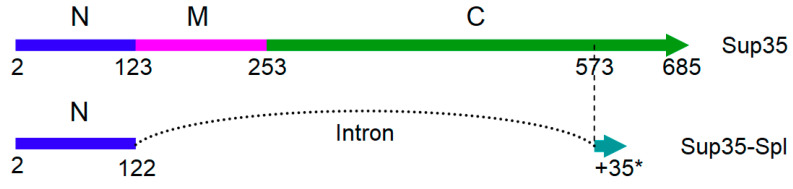
Comparison of the complete Sup35 protein and the one resulting from splicing. N, *M* and C domains of Sup35 and location of the intron are shown, along with amino acid numbering. The proteins start from residue 2, since first methionine is removed. Second exon includes 35 residues translated out of frame (*).

**Figure 6 ijms-22-01542-f006:**
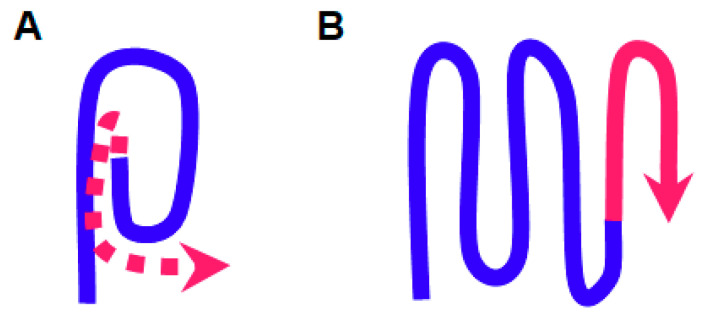
Two variants of the amyloidogenic protein terminus folding in prions and amyloids. (**A**) In the proposed buried location of the terminus (blue) it may be difficult to accommodate additional sequences (red), while no such problem should occur in the “serpentine” fold (**B**).

**Table 1 ijms-22-01542-t001:** Frequency of the [*PSI*^+^] de novo appearance upon production of Sup35 N-terminal fragments.

Sup35 Fragment	[*PSI*^+^] Appearance, 10^−5^
1–18	<1 *
1–37	<1 *
1–49	<1 *
1–64	18.76 ± 12.23
1–74	393.0 ± 222.6
1–83	1572 ± 266.0
1–93	4044 ± 313.5
1–101	7201 ± 1227
1–112	8171 ± 1613
1–123	308.5 ± 269.3
1–252	5.13 ± 1.99
Full Sup35	1.28 ± 0.08
1–101 mutant	2222 ± 857.6
1–110 mutant	1530 ± 372.2
Spliced (1–122+)	3692 ± 1504

The values show average number of [*PSI*^+^] colonies per 10^5^ cells ± standard error of the mean (SEM). The data were taken in 3–5 replicates. <1 *: no [*PSI*^+^] colonies in 10^5^ cells.

## Data Availability

No data suitable for public databases were obtained.

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
