# Peer review of "Dangerous Stops: Nonsense Mutations Can Dramatically Increase Frequency of Prion Conversion"

_ijms, 2021, doi:10.3390/ijms22041542_

Round 1

Reviewer 1 Report

Dear Editor

The manuscript by Dergalev et al. reports that truncated forms of amyloidogenic proteins originating from splicing or spontaneous nonsense mutations exhibit highly increased amyloidogenic potential. Results from this study suggest that this might play an important role in sporadic cases of amyloid diseases.

The design of the study and the technical quality of the work look convincing and results can be of general interest. The manuscript is well-written and easy to follow. Data have been presented in good way and authors used correct statistical approaches in analyzing the results. Authors have successfully managed to discuss the findings of their study through an unbiased comparison with a good range of up-to-date literature.

However, there is a number of major and minor points that would need to be addressed in order to improve the quality of this paper before it can be accepted for publication:

Major:

-It was nice to see the authors indicated some limitations of current study in towards the end of the discussion. Authors need to point out to another major limitation which is the lack of validation using humanized model where it might have an effect on post-translational modifications etc. New lines of research have discussed the variability of genetic and protein folding and profiling in 2D vs 3D. Authors need to discuss this as a limitation of the current study. Future studies can benefit from 3D self-organized models and organ-on-a-chip platforms for advanced imaging where spatial transcriptomics can be applied. References to be included:

- https://pubmed.ncbi.nlm.nih.gov/31196223/

- https://pubmed.ncbi.nlm.nih.gov/33117784/

- https://pubmed.ncbi.nlm.nih.gov/29788997/

Minor:

-It was to be way easier having line numbers.

-Abstract: Amyloid formation is the cause: this should be rephrased to “can be the trigger or causative factor or similar terms” since it’s not the only factor.

-Abstract: the second from last line: typo- “beta” seems to be missing.

-Some references are colored. Please amend.

-Introduction: typo- I.e, it should be i.e.

-Introduction “The goal of this work was to”. Was should be replaced with “is”.

-Introduction “Budding yeast Saccharomyces cerevisiae are the best-studied a”. This isn’t strictly true and needs to be amended. Maybe “one of the best-studied” or similar.

-The Y-axis of Fig2 needs to be amended to remove the underline from word auto-correction.

Best.

Reviewer 2 Report

Yeast SUP35 normally acts as a translation termination factor. However, under certain circumstances, SUP35 can form a prion (resulting in a PSI+ yeast strain). The authors use SUP35 as a model protein to examine how different classes of mutations might lead to prion formation. The significance of this study is that many prion-forming diseases in humans appear to occur sporadically and may be caused by somatic mutations that result in prion protein formation and the onset of various diseases such as Huntington's disease, Alzheimer's disease, Parkinson's disease and others.

The authors generated a yeast strain that overexpressed the prion-forming N domain of SUP35 and identified colonies that became PSI+. From 30 colonies that had become PSI+, they identified mutations within the SUP35 N domain that led to PSI+ formation. They found many of the mutations were nonsense mutations that produced a truncated SUP35 protein, with the most potent PSI+ forming mutation producing a SUP35 protein truncated at codon 112. 

To better understand the structural properties of this truncated protein, they subjected the peptide to protein kinase cleavage and found 2 core peptides that were resistance to cleavage, corroborating that this region of SUP35 was prion-forming.

Finally, the authors examined whether SUP35 was alternatively spliced in yeast, where only a small subset of transcripts are spliced. They found an alternatively spliced SUP35 transcript produced at low abundance that retained an intron, leading to premature translation termination of the SUP35 mRNA and production of a truncated protein. This truncated SUP35 strongly produced a PSI+ phenotype.

The authors suggested that the accumulation of rare transcripts containing nonsense mutations may be one mechanism of generating prion protein in somatic cells that can lead to prion diseases. 

The authors used an interesting and novel approach to examine the formation of prion formation due to the acquirement of mutations and in general, the manuscript is well-written and the data is presented in a logical way.  

However, there is often an absence of background information necessary for a reader outside of the field of yeast prions to truly understand the experimental design, the significance of the results, and whether the results support their hypothesis.

For example, on page 2, the authors describe their PSI+ screen using the ade1-14 allele that contains a premature stop codon. The authors don't describe that SUP35 is in fact, a translation termination factor and that when SUP35 is converted to its prion form, the amount of SUP35 available for translation termination is diminished, causing a suppression phenotype that results in readthrough of the ade1-14 premature stop codon, which results in a change in the color of the yeast from red to white.  Without this background information, a reader would not be able to understand this assay unless they had also performed it.

On page 4, the authors state that the Core 1 prion structure of SUP35 likely initiates prion production when full-length SUP35 is overproduced, but don't explain the reason behind this statement. This is important since Core 2 appears to be more PK resistant at the higher conc. Furthermore, the PK digestion of the N domain is not well-explained. Why were two PK concentrations used and why were 2 different digestion patterns obtained from the digests? What significance does this have on their data interpretation? For example, the authors state that no PK cleavage is observed at codon 112, yet it appears in the lower PK dose, that region of the peptide was less resistant to cleavage than the more N-terminal fragment. Overall, the interpretation of these results needs to be further clarified and their significance regarding prion structure needs to be more clearly explained in the discussion.

While the eRF3 splicing results are well-explained, why is there no diagram or figure to help the reader understand the differences between the major SUP35 transcript and the minor alternatively spliced SUP35 transcript. In addition, what is the % of the total SUP35 that is comprised by the alternatively spliced transcript. In addition, the authors indicate that "under some conditions this proportion can be significant". What conditions are the authors referring to that might increase the alternatively spliced SUP35 mRNA? What is considered to be a significant amount, in other words, how much of this transcript is normally present and how much must be present for prion formation to occur?

Finally, the discussion also contains many vague references that have not been well defined or explained. For example, the authors mention aggregation of SUP35 with Rnq1 was a first step in prion formation, but don't explain what Rnq1 is, how it fits into SUP35 prion formation, and how their data suggests this association. Furthermore, the authors don't define what IPOD is.

While overall, I think this is an interesting study that begins to delineate how SUP35 prion formation may occur, and in tern, other prion-forming proteins, the lack of clarity in the experimental design as well as the reporting and interpretation of the experimental results must be addressed before it is published.

Round 2

Reviewer 1 Report

Dear Editor,

The authors have addressed all my aforementioned comments in order to improve the quality of the manuscript. I do believe that the corrections and the additional explainations, have enhanced the clarity of the manuscript, which I endorse for publication.

All the best!

Reviewer 2 Report

The authors have adequately addressed the concerns in my previous review.